# Dendrogeomorphological Reconstruction of Rockfall Activity in a Forest Stand, in the Cozia Massif (Southern Carpathians, Romania)

Adriana-Bianca Ovreiu [1], Constantin-Răzvan Oprea [1,*], Andreea Andra-Topârceanu [1] and Radu-Daniel Pintilii [2]

1 Department of Geomorphology-Pedology-Geomatics, Faculty of Geography, University of Bucharest, 010041 Bucharest, Romania; adriana-bianca.ovreiu@geo.unibuc.ro (A.-B.O.); andreea.andra@geo.unibuc.ro (A.A.-T.)

2 Research Center for Integrated Analysis and Territorial Management, Department of Human and Economic Geography, Faculty of Geography, University of Bucharest, 010041 Bucharest, Romania; radu.pintilii@geo.unibuc.ro

* Correspondence: constantin.oprea@geo.unibuc.ro

**Abstract:** Determining the spatio-temporal patterns of rockfalls, such as the zonation of hazards and the assessment of associated risks, can be challenging due to poor historical archives. Dendrogeomorphological methods cover this lack of data and provide reliable reconstructions of rockfall activities over several centuries. These methods are based on the signals recorded in the tree rings that are affected by the mechanical impact of falling rock fragments. In this study, we analyzed the spatial and temporal distribution of rockfalls in a 0.19 ha forest area in the Southern Carpathians. We collected 170 samples (100 increment cores and 70 stem discs) from all 40 *Picea abies* (L.) Karst trees identified in the study area (1 tree/47 m$^2$). This allowed us to date 945 events between 1817 and 2021, which we then compared with available weather records. Our results show the main trajectory of falling rock fragments from the source area, as well as significant temporal variations in process activity. These variations correlate only slightly with fluctuations in meteorological parameters. Despite the expected intensification of natural hazards due to climate warming, our study area shows a general trend towards a slight decrease in rockfall activity at present.

**Keywords:** dendrochronology; rockfall events; spatio-temporal reconstruction; forest stand dendrogeomorphic potential; Southern Carpathian

## 1. Introduction

A rockfall is a natural hazard that can affect human-made infrastructure and activities in mountainous regions. It occurs when a block or several rock fragments move down a slope in different ways, e.g., by free falling, bouncing, rolling, or sliding in the area of material accumulation [1].

Due to the increasing frequency and intensity of natural disasters caused by climate change, rockfall processes pose a growing risk on slopes [2].

These geomorphological processes exhibit considerable spatio-temporal variations, leading to difficulties in predicting events [3]. Monitoring rockfalls and identifying patterns of occurrence are crucial to reducing or even eliminating the human and material damage caused by these slope processes [4].

According to [1], there are three categories of methods applied to assess the magnitude of rockfall activity: (a) reconstructing a timeline of past rockfall events using historical records of rockfalls [5,6] or various quaternary dating methods [7–9]; (b) monitoring of steep slopes and inventorying new rockfall events by in situ observations and measurements [10] or by remote sensing techniques [11–13]; or (c) estimating the frequency and magnitude of events by modeling the distribution of rockfall volumes [14,15].

Methods for dating past rockfall events provide long inventories, on the order of centuries or even millennia when historical archives are missing or are incomplete. Although they have a lower resolution of dating compared to other types of approaches, these methods can provide important information about the spatio-temporal patterns of rockfall activities [1].

The present study aims to reconstruct past rockfall events using dendrogeomorphological methods by analyzing the reactions of tree growth rings to mechanical impacts [16]. Dating the activity of rockfall events by studying the tree growth rings is widely used and has been described in numerous studies. A comprehensive overview of dendrogeomorphological studies was conducted by [9]. More recent approaches focus on the assessment of the spatial and temporal distributions of rockfall frequency in correlation with meteorological parameters and climate change [17–20].

This paper proposes two directions of approach to study rockfall activities: (a) the temporal reconstruction of the rockfall events over the last two centuries, the identification of the years with the highest rockfall activity and their correlation with meteorological parameters to quantify the influence of climate warming on the trend of process dynamics, and (b) the spatial reconstruction of the rockfall, which leads to hazard zonation and a delineation of the main trajectory of the rockfall and can provide reliable input data for research on risk management and mountain spatial planning.

## 2. Materials and Methods

### 2.1. Study Area

The study area is in the northern side of the Cozia Massif, in the Southern Carpathians, Romania and depicted in Figure 1a. A steep slope made of gneiss is the source of the rockfall, located at an altitude of approximately 1400 m and approximately 40 m elevation. This slope faces south (Figure 1b,c). The gneiss's mineralogical heterogeneity and microtectogenic deformations make it more susceptible to weathering processes and release decimeter-sized rock fragments. The analyzed site lies in a temperate zone, which leads to rockfall occurrences during the transition seasons when freeze–thaw cycles could frequently occur daily. During these periods, the thermal weathering of the rocks is the most intense and often is enhanced by precipitation. According to the ROCADA's weather data (1961–2013) [21], the site's average annual temperature is 4.17 °C, and the total precipitation is 915 mm/year.

The slope of the rock fragments that break away from the source area has a transit and accumulation profile with an inclination angle ranging from 25° to 65°. The slope surface is divided into two thresholds based on the size of the rock fragments. The upper sector, near the source's steep slope, has fragments smaller than 0.15 m in diameter, while the lower sector has fragments of approximately 0.7 m in diameter. Additionally, larger clasts likely remnants of large-scale events from the past were identified below the second threshold (Figure 1c,d). The dominant geomorphological process that occurs on the slope is given by rockfall activity, and no signs of other gravitational geomorphological processes causing tree damage have been identified.

The investigated stand forest occupies the upper half of the talus and extends about 50 m from the base of the source area to an altitude of 1350 m. Covering an area of 0.19 hectares, the sampling stand is a mixed forest of *Picea abies* (L.) Karst and *Fagus sylvatica* (L.) trees. The site is a protected area within the Cozia National Park and has not been subjected to any forestry exploitations, also due to its high slope values and limited accessibility. While there have been no significant tree mortality periods, there may have been limited forestry or biological interventions in the past century, although there is no evidence to support this. Currently, anthropogenic activities are restricted to the tourist path and do not affect the reconstruction of rockfall events. However, the lack of historical archives on rockfall events in the area from the past is due to the minimal human interference in the study site.

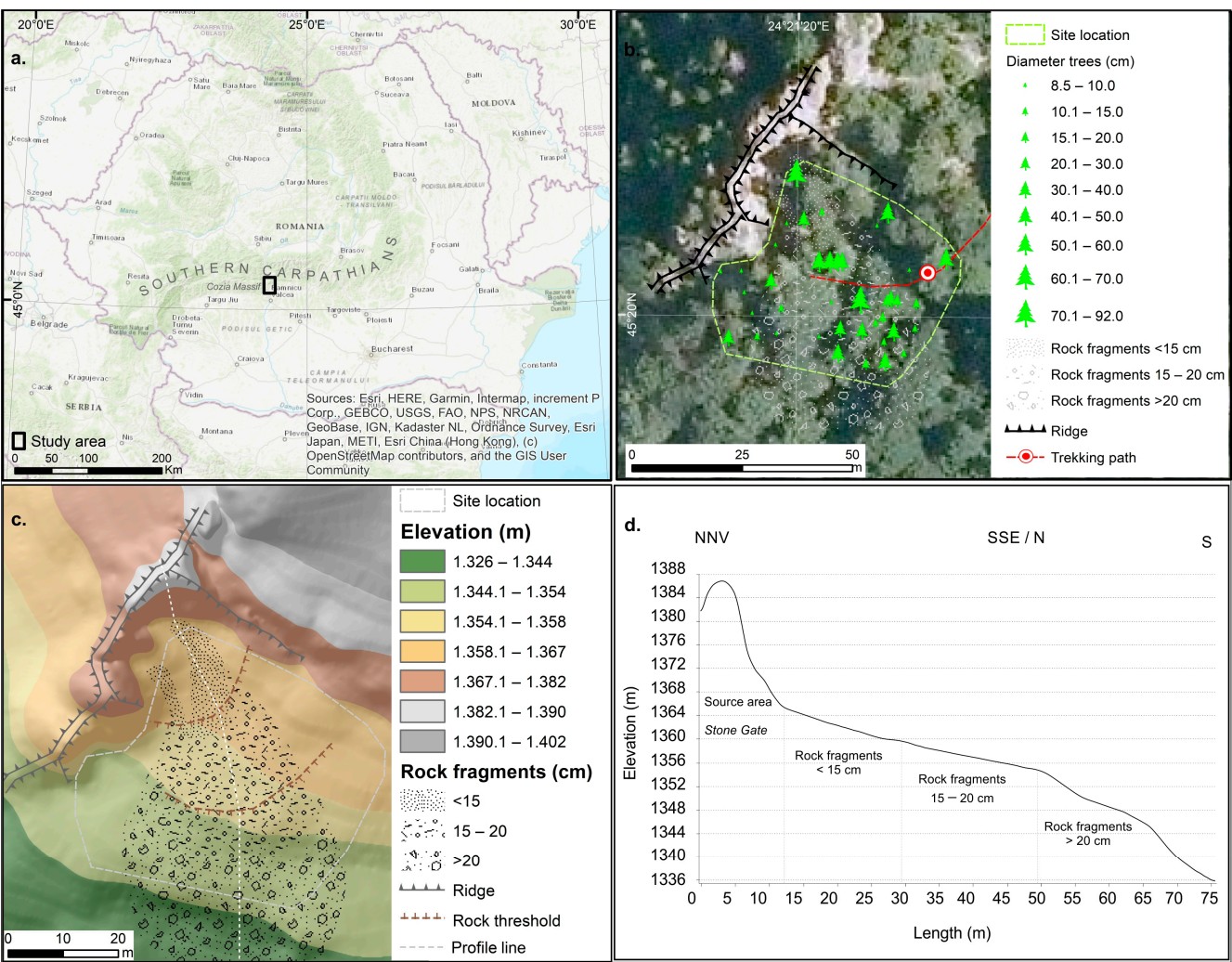

**Figure 1.** The Study area (**a**) within Romania and the Southern Carpathians, (**b**) the site with the positions of the sampled trees, (**c**) distribution map indicating the location of the rockfall fragments and the longitudinal geomorphological profile line along the rock fragments accumulation area, (**d**) the slope geomorphological cross-section of the transit and accumulation of released rock fragments.

The tourist route crosses the slope that is prone to frequent rockfalls, thereby posing a potentially grave risk to tourists. In light of this context, it is imperative to conduct a detailed analysis of the spatio-temporal distribution models of rockfall patterns to ensure proper planning and management of the mountainous area.

### 2.2. Dendrogeomorphic Sampling and Analysis

Our approach is based on data processed from 170 dendrochronological samples, including 100 increment cores and 70 stem discs taken from 40 *Picea abies* trees. All coniferous trees in the study area were included in the analysis, without bias or exclusion of any individual tree. We followed the methodology described in previous studies [20,22,23], which involved sampling the entire width of the slope, but only descending 50 m down the slope relative to the source area, to eliminate the errors induced by the multiple impacts made by the same rock block. These studies consider that the optimal sample size of 40-80 trees/ha should provide reasonable accuracy to dendrogeomorphic reconstructions. In our approach, we investigated 40 trees, which is the minimum necessary stipulated from previous works but distributed over only 20% of the minimum area required (0.19 ha representing 1 tree/47 sqm). We did not identify any periods of forest exploitation or significant tree mortality [24]. Regardless of the forest stand's historical evolution, we

ensured that the sample's depth met the literature's recommended conditions during the analysis.

To construct a local reference timeline that depicts the typical variation in tree growth within the study area, an additional group of 15 *Picea abies* was sampled from a nearby forest that had not been disturbed by geomorphological processes such as rockfalls [25,26].

The standard procedures outlined by Mainieri et al. (2019), Stoffel et al. (2010), Trappmann and Stoffel (2013), and Braker (2002) [9,16,23,27] were followed for the extraction of increment cores using a Pressler drill, removal of stem discs with a diameter of less than 15 cm, and processing of the samples. The counting of growth rings and their width measurements were conducted in the dendrochronological laboratory using a digital positioning table LINTAB connected to a Leica stereomicroscope and TSAPWinTM software [28].

Past rockfall events were identified from 1817 to 2021 based on the tree growth response to cambium damage as evidence of the mechanical impact of the rockfalls (Figure 2). Various growth disturbances (GDs), such as impact scars (SCs), traumatic resin ducts (TRDs) that form continuous, compact, and tangential rows, compression wood (CW), and sudden suppression of growth rings (GSs), indicated the occurrence of those past rockfall events [29–31].

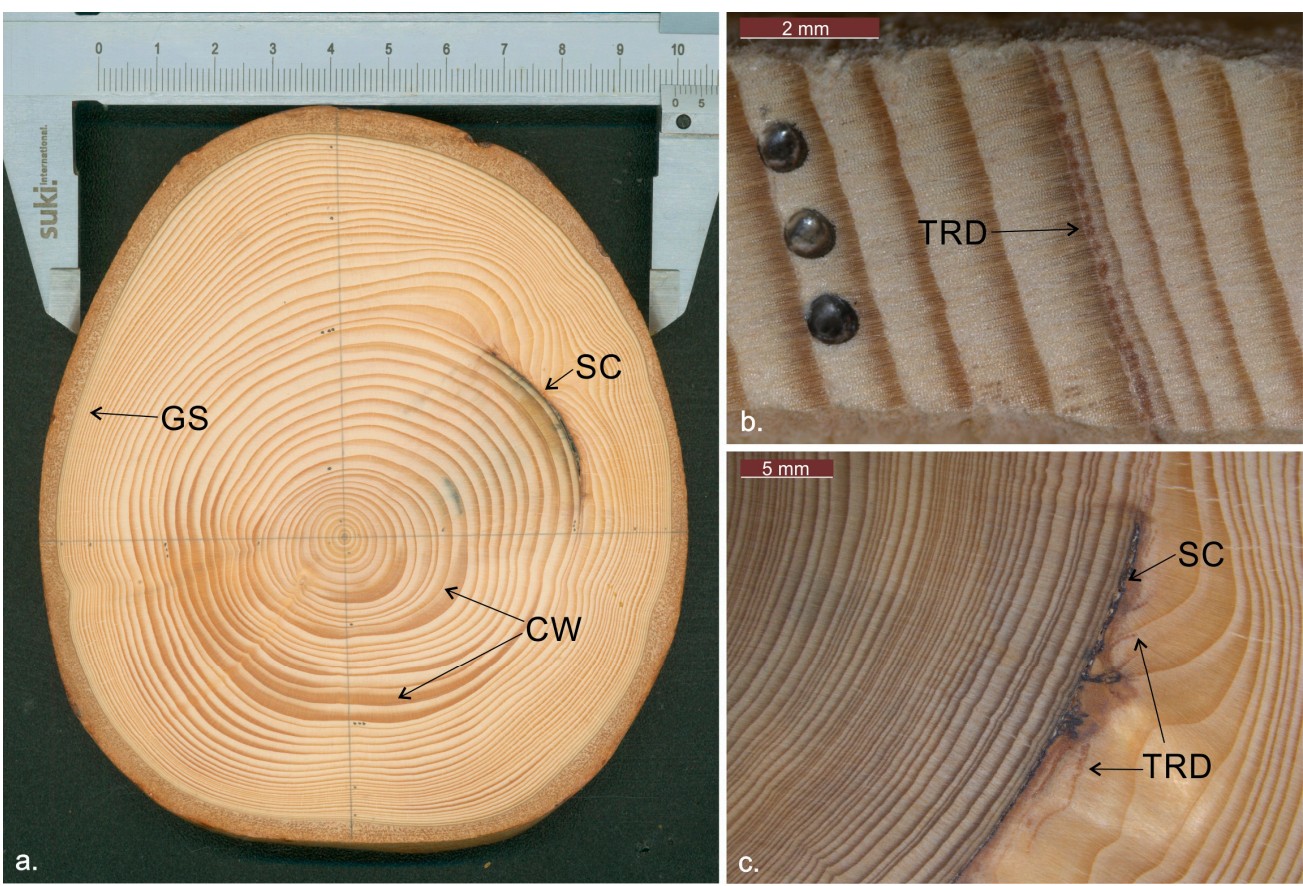

**Figure 2.** Growing disturbances (GDs) generated by rockfall events identified through sampled *Picea abies* (L.) Karst growth rings: (**a**) SCs (impact scars), GSs (sudden suppression of growth rings), and CW (compression wood), (**b**) TRDs (traumatic resin ducts), (**c**) SCs and TRDs.

Since an impact could generate more anatomical responses in growth rings, a single rockfall event per year was dated for each individual tree. Also excluded from the analysis were the identified growth disturbances in the first 10 years of life of each impacted tree, given the high susceptibility of young trees to environmental factors. Therefore, this procedure is considered more accurate and may provide at least the minimum number of identified rockfall events [32–34].

The temporal analysis of rockfall activity was based on calculating a metric called conditional impact probability (CIP), which determines the area covered by trees each year and estimates any potential rockfall events that may have missed the tree stems [35,36]. Each tree covers a slope range exposed to rockfalls, defined as an "impact circle" that determines its probability of being hit by a rockfall. This "circle" is given by the diameter of the tree, known as the DBH, measured at 130 cm height and the average diameter of the released rock fragments. The amount of all "impact circles" represents the area covered by trees each year [20].

The calculation of CIP requires data about the location of the trees, their measured DBH for each tree, and the average diameter of the fallen rock fragments (25 cm in our study). The formula used to calculate CIP is CIP = LIC/Lplot, where LIC is the total diameter of the impact circles or the slope range covered by the trees, and Lplot is the width of the analyzed slope. In our study, Lplot was 45 m. These findings were based on previous studies conducted by Moya et al. in 2010 [35] and Favillier et al. in 2017 [36].

Then, the CIP values serve to apply corrections for a more accurate reconstruction of rockfalls and to estimate the actual or "real" annual number of rockfall events (RRs) in a given year, as follows: $RR_t = NGD_t/CIP_t$, where $NGD_t$ is the number of growth disturbances dated in year $t$, and $CIP_t$ is the conditional impact probability calculated for the same year $t$ [17]. If no growth disturbance is recorded in the tree rings, then the RR value will be 0. As the CIP values decrease, the RR values will increase, highlighting the spatial patterns of rockfall activity [20].

To highlight the spatial dynamic patterns of rockfall activity over the investigated period (1817–2021), the number of GDs recorded per tree was illustrated through ArcGIS interpolation. The data from tree ring analysis were divided into 5 time separated sequences, 4 time sequences of 30 years each for the range of 1900–2021 and 1 single sequence for the period of 1817–2021 due to the very small sample size remaining until nowadays (only 4 trees available for reconstruction).

To summarize the meteorological context with the potential for triggering rockfalls, air and soil temperature data (annual, monthly, and seasonal averages, maximum, and minimum values) and precipitation (annual and seasonal totals) were extracted from the ROCADA climate database (1961–2013) [21] and then correlated with the reconstructed activity of rockfalls in the study area. The relationship between the rockfall activities and meteorological parameters was quantified by calculating the Pearson correlation coefficient between RR and the weather variables and statistically tested by determining the coefficient *p* at a significance level of 0.05 [37].

## 3. Results

In this study, the sampled trees range from 31 to 215 years, with an average age of 104 years. The sampling period spanned from 1807 to 1991, with the oldest tree reaching sampling age in 1807 and the youngest in 1991. The average diameter of the trees examined was 26.85 cm, with a minimum of 8.5 cm and a maximum of 92 cm. The growth rings of the trees revealed 1137 GDs caused by rockfalls, and these were further categorized into 457 GSs (40%), 431 CWs (38%), 206 TRDs (18%), and 43 SCs (4%) [24]. We noticed that some of the 1137 GDs recorded are simultaneously located in the same tree's growth ring or were recorded during the first 10 years of the tree's life. Due to GDs simultaneity having the potential to induce errors in the reconstituted rockfall activity by overestimation, we removed them from the dendrogeomorphological analysis. After dendrochronological dating and from the GDs total, 945 growth disturbances were left, which correspond to as many rockfall events (data are available in the Supplementary Materials).

The temporal analysis of rockfall activity in the study area began in 1817 when the oldest tree reached the sampling age of 10 years. The sample size increased sharply until 1931 when 28 trees (70%) with ages over 10 years were available for sampling. Starting from the early 1910s, 25% of the trees (>10 years) exceeded the sampling height. By contrast, the slope range covered (LIC) by the trees showed a gradual increase from 0.14 m at the

beginning of the analyzed period to a maximum of 16.15 m in 2021 at the end. We noticed that there was a relatively steep increase in LIC between 1892 and 1931 when the number of trees increased by seven times (Figure 3a).

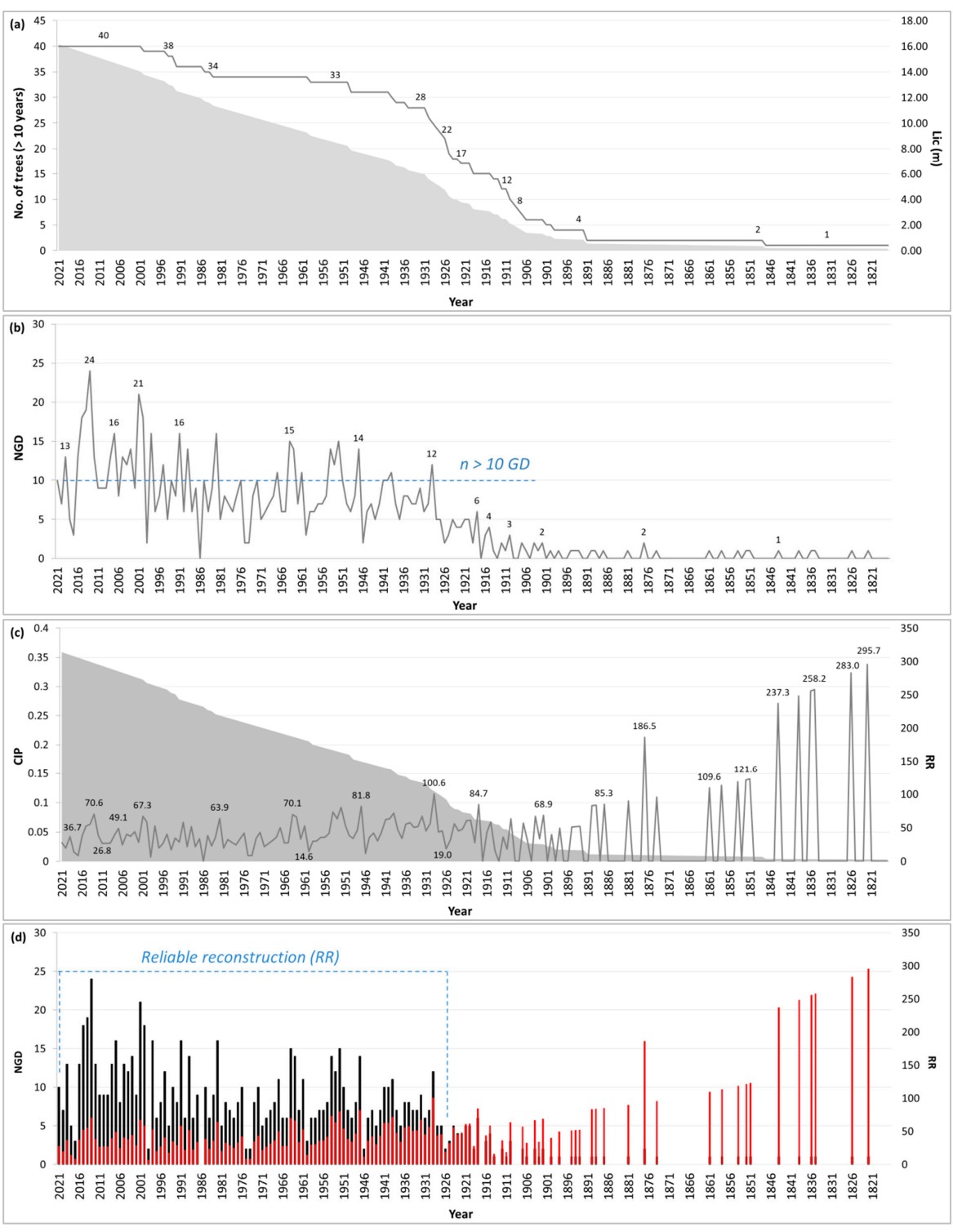

**Figure 3.** The sample features and reconstruction of rockfall activity: (**a**) number of trees (>10 years) and tree-covered area (LIC, shaded area); (**b**) absolute numbers of rockfall events (NGDs); (**c**) conditional impact probability (CIP, shaded area) and "real" number of rockfalls (RR); (**d**) absolute numbers of rockfall events (NGDs, black bars) and number of rockfall events corrected with CIP (RR, red bars).

The activity reconstructed by rockfalls points out an upward general trend over the analyzed period, surpassing for the first time the threshold of 10 GDs/year [20] in 1929 (12 GD). In the time frame of 1929–2021, there were 845 GDs (89.41%) of the total of 945 GDs identified for the period of 1817–2021. During this time frame (1929–2021), in 36 of the years, the values reached or even exceeded the threshold of 10 GDs/year, totaling 490 GD, which is over half (51.85%) of the total recorded for more than two centuries (205 years) (Figure 3b, Table 1).

**Table 1.** Rockfall activity reconstruction was determined through the dating of tree growth disturbances (NGDs), which are over the 10 GDs/year threshold and CIP corrected chronology (RR) values of more than 100 rockfall events.

| NGD (*n* > 10 GD) | | | | | | | | | | | |
|---|---|---|---|---|---|---|---|---|---|---|---|
| Values | 24 | 21 | 19 | 18 | 16 | 15 | 14 | 13 | 12 | 11 | 10 |
| years | 2013 | 2001 | 2014 | 2000 2015 | 1982 1991 1998 2007 | 1952 1964 | 1947 1954 1963 1989 2003 | 2005 2008 2012 2016 2019 | 1929 1953 1995 2004 | 1939 1961 1967 | 1940 1941 1951 1972 1976 1985 1993 2021 |

| RR (*n* >100) | | | | | | | | | | | | |
|---|---|---|---|---|---|---|---|---|---|---|---|---|
| values | 295.7 | 283 | 258.2 | 255.7 | 248.5 | 237.3 | 186.5 | 123.1 | 121.6 | 118.7 | 113.3 | 109.5 |
| years | 1822 | 1826 | 1835 | 1836 | 1839 | 1844 | 1877 | 1851 | 1852 | 1854 | 1858 | 1861 |

The average CIP result is 0.122 (12.2%), with the highest value being 0.3589 (35.9%) and the lowest being 0.0032 (0.32%). The CIP shows a positive trend, with higher values and increased evolution in the recently analyzed time intervals when the slope had the most forest stand cover, and also a higher number of growth disturbances in the ring series was recorded compared to lower values in the distant past. The CIP was applied to correct the reconstruction, taking into account rockfalls that did not generate growth responses in tree rings (Figure 3c).

Upon correcting the data with CIP, a downward trend in rockfall activity is observed, with significantly higher values recorded in the distant past and decreasing values for the recent periods of reconstruction. The HR results indicate an average of 37.91%, with a maximum of 295.7 recorded in 1822 when the CIP had the lowest values and a minimum of 6.6 in 1999 when CIP was nearing its maximum value. Furthermore, RR values of zero were also recorded for 69 years when no GDs were identified in tree rings. The highest RR values were observed for some years from the XIX century when the maximum (295.7) of the entire analyzed period was recorded in 1822 (Table 1). We noticed a decline in the rockfall activity, after numerous rockfall events which had occurred from 1821 until 1877, the RR values being at a comparable level for the entire time sequence between the end of the XIX century and the present. In this interval, the most significant activity of rockfalls was recorded in 1929 when the RR value was 100.6. In the XXI century, the RR values oscillate between 8.6 (2017) and 70.6 (2013), with an average of 37.85, similar to the calculated mean value for the entire analyzed interval. After 2017, there was a steep increase in the number of events until 2019 (R = 36.7), followed by a decrease in the rockfall activity by 53.5% in 2020 (R = 19.63). Although the value of the RR (27.9) increased by 1.4 times in 2021 compared to the previous year, the general trend of the activity of rockfalls is currently slightly decreasing (Figure 3d).

The rockfall frequency for individual trees was calculated and their spatial distribution was determined across different time sequences, as well as for the entire studied interval of 1817–2021 (Figure 4). The average number of rockfall events dated per tree is 23.6, the

maximum value is 63 events, and the 3 events per sampled tree are the minimum values. Trees on the southeastern part of the slope recorded the highest number of events, with one tree registering the maximum number of 63 impacts. The north-central sector of the slope, near the source area, had medium values (20–30 events/tree), with the trees registering predominantly more than 25 events/tree. However, just some areas close to the rock wall source are characterized by more than 30 impacts/tree. Rockfall activity was less frequent (<20 events/tree) on the downslope farthest from the rock fragments' source area and near the steep threshold that delimits the western slope, where most trees register between three and eight events. This fact may be explained by the very low age of the trees that limit the reconstruction of rockfall events but especially by the low tree diameters that constituted a shallow barrier to rockfalls; most blocks are down in the central–southeast of the slope where the trees are frequently disturbed.

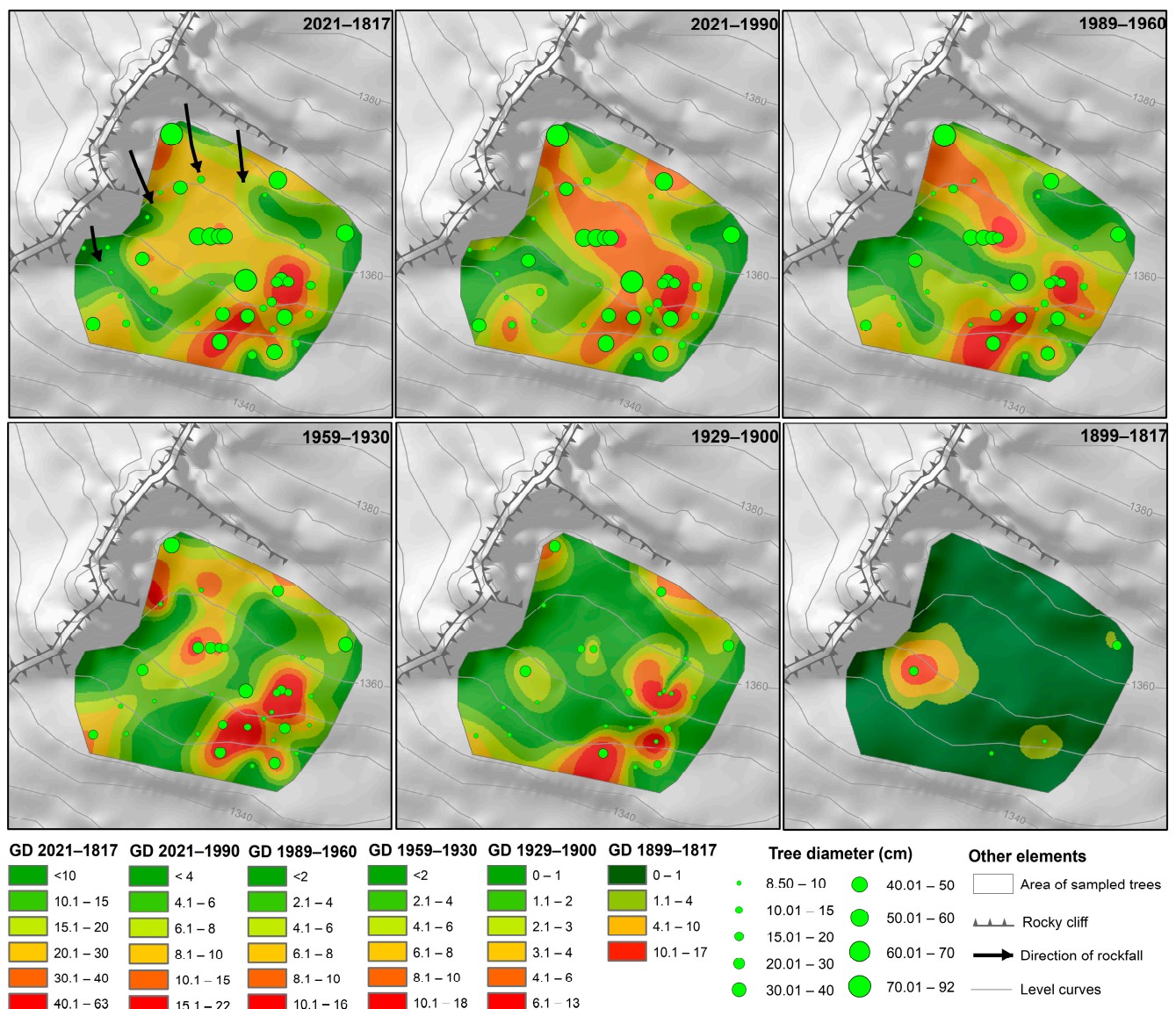

**Figure 4.** The pattern of spatial distribution of the rockfall activity within the survey area has been reconstructed through the interpolation of the growth disturbances (GDs) in tree rings for different time sequences.

The most accurate representation of the reconstructed spatial distribution of rockfall activity from the past two centuries is the map for the 1990–2021 time sequence that reveals

the main trajectory of rock falling. Based on the tree's growth rings that recorded up to 22 rockfall events, the map shows that the rock fragments have been released from the north-northwest detachment area where gravitational process trajectory is towards the center and southeast of the slope. Along the described trajectory, the main block-fall corridor started forming in the early 1930s when trees clustered on the NNV-SSE direction of the slope and recorded up to 18 rockfall events. The time sequence between 1900 and 1929 highlights the impact of sample size, exposed tree diameter, and distance between trees on the spatial and temporal reconstruction of rockfalls. Young, small-diameter, and spaced apart trees are less likely to be impacted, and most rock fragments left growth reactions in tree rings located in the southeast sector of the slope. The map reflecting rockfall activity between 1817 and 1899 shows that very small samples from the distant past of the reconstruction can lead to erroneous results on the manifestation of the process during these periods (Figure 4).

## 4. Discussion

Relying only on the absolute number of GDs dated in the tree growth rings to reconstruct rockfall activity could lead to an overestimation of current events. This approach reflects the growth trend of the sample, the tree diameters, and the degree of slope cover, rather than the frequency of rockfalls from recent reconstruction periods. The CIP-corrected reconstruction (RR) addresses some of these limitations by taking into account the temporal changes in the sample and tree-covered area (LIC), which represents the probability of impact. However, this method remains sensitive to very small samples from the distant past of the dendrogeonorphological reconstruction, leading to very high values of rockfall activity for this period.

Therefore, the reconstructions presented in this study are reliable for the results of the dendrogeomorphological analysis for the time interval between 1910 and 2021 if we consider the minimum threshold of 25% from the sample, as suggested by Šilhán et al. in 2013 [25] and a threshold of only 5% of CIP. Therefore, this will lead us to limit the time range to 1926–2021, as reliable interval, when the CIP recording value of 10.5% [20], due to a minimum of 22 trees were already 10 years old and were available for sampling (55% of the sample). In the time interval of 1926–2021, for which the dendrogeomorphological analysis results can be considered reliable, the rockfall activity shows significant spatio-temporal variations, partially correlated with the variations in the meteorological parameters.

The correlation between the CIP-corrected rockfall reconstruction and meteorological parameters was analyzed using data from the ROCADA datasets. The correlation coefficients and statistical significance were calculated and shown in Table 2.

**Table 2.** Correlation (R) and statistical significance (*p*) coefficients were calculated for the reconstruction of the CIP-corrected rockfall activity (RR) and meteorological parameters.

| Coefficients | R | *p* |
|---|---|---|
| **Variable** | **RR** | |
| Annual average temperature | 0.090 | 0.52 |
| Annual average minimum temperature | 0.060 | 0.67 |
| Annual average maximum temperature | 0.011 | 0.94 |
| Mean annual soil temperatures | 0.014 | 0.92 |
| Total precipitation | 0.017 | 0.90 |
| March–May Temperature | 0.070 | 0.62 |
| March–May Precipitation | 0.010 | 0.94 |
| June–August Temperature | **0.230**[1] | 0.10 |
| June soil Temperature | **0.300**[1] | **0.03**[1] |
| June–August Precipitation | 0.150 | 0.28 |
| September–November Temperature | **0.200**[1] | 0.15 |
| October Temperature | **0.300**[1] | **0.03**[1] |
| September–November Precipitation | **0.300**[1] | **0.03**[1] |
| December–February Temperature | **0.300**[1] | **0.03**[1] |
| December–February Precipitation | 0.140 | 0.32 |

[1] Bold numbers indicate significant statistical correlations.

According to the coefficients, climatic factors do not appear to have a significant influence on rockfall activity. We found no significant correlation (R > 0.5, *p* < 0.05) between the calculated timelines and the following meteorological parameters: annual average temperature (R = 0.09, *p* = 0.52), annual average minimum and maximum temperatures (R = 0.06, *p* = 0.67 and R = 0.01, *p* = 0.94), annual total precipitation (R = 0.017, *p* = 0.9), average temperature and total precipitation for the spring season (R = 0.07, *p* = 0.62 and R = 0.01, *p* = 0.94), total precipitation in summer (R = 0.15, *p* = 0.28), and total precipitation in winter (R = 0.14, *p* = 0.32).

The average temperature of June-August had a weak, statistically insignificant correlation (R = 0.23, *p* = 0.1).

The study revealed weak but statistically significant correlations (R = 0.3, *p* = 0.03) between the amount of precipitation and the average temperature in October, the average winter temperatures (December–February), and the average soil temperature recorded in June (Figure 5).

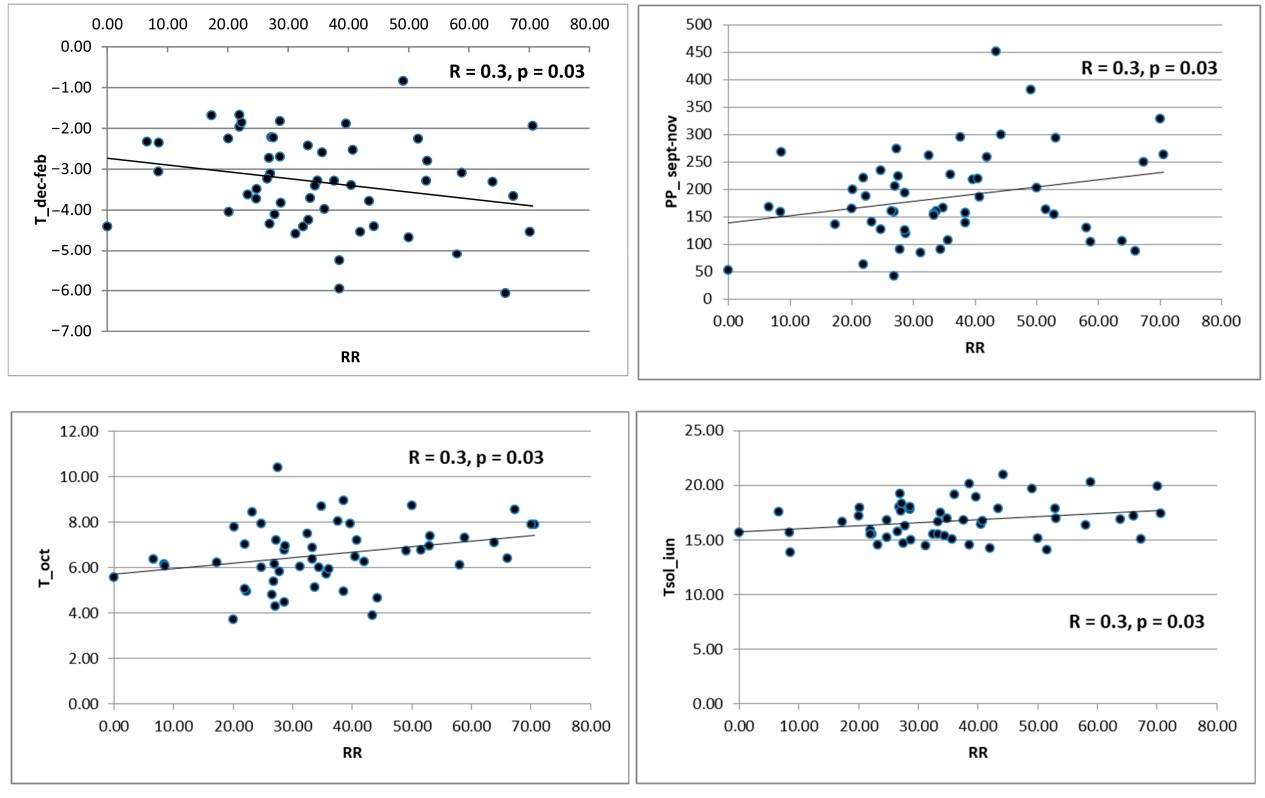

**Figure 5.** Pearson's correlation between reconstructed rockfall activity (RR) and meteorological parameters (average temperature months December–February, total rainfall for September–November, average temperature of October, and average soil temperature in June).

These correlations could be due to the triggering of a geomorphic process under the influence of local heavy rainfalls in the autumn months, as a result of an increase in pressure in the crack system of the blocks. The high-frequency release of rock fragments may also result from multiple freeze–thaw cycles in winter or in the transition seasons, which lead to the growth of the "stress thermal" of the rocks [38] and the development of pre-existing cracks. During the summer months, especially at the beginning of the growing season when the slopes are not yet so heavily vegetated, rockfalls can occur due to disaggregation caused by solar radiation, especially on the south-facing rock slopes. These rock walls are composed of rocks and minerals with different thermal and physical properties that are exposed to significant temperature variations at altitudes [39].

The global trend of climate warming may lead to an increase in the frequency of rockfalls due to more frequent freeze–thaw cycles, melting snow and permafrost, temperature changes, heavy rainfall, and local earthquakes, which are the main triggers of this process [17]. However, several studies [17,18,20,40,41] have reported that there are no clear correlations between the reconstructed rockfall activity and fluctuations in the meteorological parameters. Sass and Oberlechner (2012) [40] believe that there is no general upward trend in the frequency of rockfalls in observational records that is related to variations in climatic factors. Šilhán et al. (2012) [41] state that no significant correlation between rockfall and precipitation was identified. Similarly, Mainieri et al. (2021) [20] found no statistically tested correlation between the dated events and temperature (average, maximum, minimum) or the freeze–thaw cycles. On the other hand, Perret et al. (2006) [33] showed that the rate of rockfalls correlates to a high level of statistical significance with the average annual temperature, as well as summer and winter ones. Macciotta et al. (2015) and Matsuoka (2019) [42,43] reported an increased frequency of rockfalls following episodes of heavy rainfall that increases pressure in cracks and favors meteorization.

The discrepancies in the results of various studies addressing the relationships between the rockfall activity and the thermal meteorological variables can be explained by several factors:

(a) The dendrogeomorphological records have a limited resolution and may not capture common events with low magnitudes [43];

(b) The dating of growth responses due to rockfalls is accurate only to an annual level, rather than seasonal or sub-seasonal [44];

(c) The altitudes of the study areas vary from one study to another, and the influence of climate warming on rockfall activity is observed predominantly at high altitudes, related to glacial equilibrium-line altitude or where permafrost occurs. Below the permafrost limit, the dependence of rockfall activity on climate variability remains poorly understood, although most events posing a threat to anthropogenic infrastructures and activities come from slopes well below this limit [17];

(d) The climate dataset does not capture microclimatic variations and the effect of the abrupt microtopography [18];

(e) The complex conditions of manifestation of frost–thaw related to the degree of humidity and heating periods [20].

To obtain a more detailed analysis of the relationship between climate change and rockfall activity, the following steps are necessary:

(a) Investigate a large number of slopes to obtain rockfall dates and chronologies that are not influenced by local factors. This can provide insight into climate trends affecting rockfall activity [17];

(b) Analyze daily historical records of rockfall activity and meteorological parameters to determine the degree of correlation between these variables [20];

(c) Narrow the downward slope of the sampling area while expanding its width [23];

(d) Systematically sample the first rows of trees near the source area, provided that the trees are old enough to allow for dendrogeomorphological reconstructions over large time intervals [45].

The growth rings of trees can register environmental changes, such as climate and geomorphological processes. These growth disturbances allow us to reconstruct rockfall activity over long periods. Our study highlights the potential of trees both for the reconstruction of rockfalls and the evolution of tree characteristics, such as number of trees, diameter, age, density, and spatial distribution.

In Figure 4, the spatial pattern of rockfalls is shown alongside the evolution of the tree features. The number of trees that have reached at least the sampling height, which is 1.30 m, has increased unevenly from four trees in 1899 to 40 in 2021, with an average rate of three trees per 10 years. The maximum variation number was up to 25 trees in 1929, and the current number of trees (40) was reached in 1990. The structure of the forest stand in its

current shape (number of trees, diameter, spatial distribution, including tree clusters) was completed in the 1960s. Subsequently, major changes only occurred in the diameters of the trees. Therefore, the same database obtained in the field can be used for both analyses of geomorphic processes and the evolution of forest landscapes.

## 5. Conclusions

Forested slopes facilitate the reconstruction of rockfall activity by analyzing growth disturbances imprinted in the tree rings. In the studied forest stand, the most common anatomical responses to the rock fragments' impacts are sudden growth suppression, compression wood, traumatic resin ducts, and impact scars. The dating of growth reactions on increment cores and stem discs led to a spatial model of the number of rockfall events. However, the frequency of the process was only determined after correcting the data with CIP, which takes into account the temporal variation in the sample and the range of the covered slope. After the rectification, a significant decrease in rockfall activity was observed in the recent decades, compared to the distant periods of reconstruction when much higher values of rockfall activity were recorded.

The increase in rockfall activity is only slightly related to meteorological parameters, such as autumn precipitation, average temperature of October, average winter temperatures, and average soil temperature recorded in June. However, there is no correlation between the calculated timelines and the annual and seasonal average temperature or the annual and seasonal rainfall totals. To better understand the role of meteorological factors in rockfall fluctuation (variation), microclimatic monitoring of the site, improving sampling methods, and increasing the resolution of events dating from annual to seasonal or sub-seasonal are required. We aim to address these issues in a future paper.

Despite this, the results of the present analysis confirm the potential of trees and dendrogeomorphological methods for the spatio-temporal reconstruction of rockfalls. This is especially useful in regions where there are no historical records or where the inventories of rockfalls are incomplete and cover short time intervals. These results can be helpful to decision makers who can use them as input data in hazard zoning at rockfalls and analysis of associated risks.

**Supplementary Materials:** The following are available online at https://www.mdpi.com/article/10.3390/f15010122/s1.

**Author Contributions:** Conceptualization and writing A.-B.O. and A.A.-T.; methodology, A.-B.O.; laboratory and software, A.-B.O. and A.A.-T.; validation, A.-B.O., C.-R.O. and A.A-T.; analysis, A.-B.O., A.A-T. and R.-D.P.; investigation, A.-B.O. and A.A-T.; data curation, A.-B.O., R.-D.P. and C.-R.O.; writing—review and editing, A.-B.O., A.A-T. and R.-D.P.; funding acquisition, A.-B.O. All authors have contributed equally to this manuscript. All authors have read and agreed to the published version of the manuscript.

**Funding:** This research was funded by the RESEARCH INSTITUTE OF THE UNIVERSITY OF BUCHAREST (ICUB), grant TC ICUB number 2148/01.02.2022.

**Data Availability Statement:** All data are available in the Supplementary Materials.

**Acknowledgments:** The authors would like to thank to the Cozia National Park staff.

**Conflicts of Interest:** The authors declare no conflicts of interest.

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
