# Peer review of "Dendrogeomorphological Reconstruction of Rockfall Activity in a Forest Stand, in the Cozia Massif (Southern Carpathians, Romania)"

_forests, doi:10.3390/f15010122_

Round 1

Reviewer 1 Report

Comments and Suggestions for Authors

Dear authors,

This is an interesting paper concerning the rockfall activity and the usefulness of dendrochronology in its temporal and spatial reconstruction. I have some general comments that you may find below, together with some minor comments and edits which could possibly be useful for you.

General comments

1) In the abstract (lines 17-8) it is written: “from all 40 Picea abies (L.) Karst trees identified in the study area”. In Fig. 1b only trees with diameter higher than 8.5-10 cm are illustrated? Does this mean that you measured/ took into account only trees with D>8.5 or you did not have younger trees in the selected study site? Please clarify.

2) Part of the discussion section where the correlations with climate are presented should move to the Results Section.

3) In different parts of the manuscript you use the term “global climate change”. Global is not necessary. “Climate change” is enough.

4) Sometimes you use rockfall(s) as one word and others as two words. Please homogenize.

5) I think a better connection with the scope of the special issue is needed. You should try to emphasize the usefulness of your findings in terms of the landscape protection and/or the provided ecosystem services.

Specific comments

Line 17: identified or recorded?

Line 28, lines 33-34: “to human-made infrastructures and activities in mountainous regions”. There is some repetition here. Please revise. Also, why to mention only the human made infrastructure? It also affects the natural environment.

Line 46: “or incomplete”. Better to replace it with “or are incomplete”

Line 54: the parenthesis is not necessary

Lines 57: I do not understand what do you mean with “two directions of approach”. By reading a) and b) I would write something like this: “rockfall activity was studied in two directions..”

Line 74: “by precipitations which turn into rain or snow.” Otherwise what are the precipitations?

Line 88: Give the full scientific name for Fagus sylavtica as you did for Picea.

Line 94: delete double “do not”

Lines 108, 122 etc: no need to repeat the full name. Please replace just with Picea abies. It is enough to give the full scientific name once at the beginning

Figure 2: Explain in the figure caption all the abbreviations appearing in the figure. Figures should be self-explicit.

Lines 188-189: “into 457 were GS…” English revision is needed. All the “were” are not necessary.

Lines 189192: English revision is needed. The meaning is unclear.

Line 262: “within survey area”. Add “the” before survey.

Lines 276-279: This is a common problem when dealing with the reconstruction of past events/ disturbances, (e.g. fire history reconstruction etc). I would like to see a short paragraph about this in the Discussion section.

Line 291: “for time interval”. Add “the”

Line 293: “we taking in account”. English revision is needed

Line 368: are necessary.

Comments on the Quality of English Language

Only minor English editing is required

Author Response

We, the authors of this manuscript, express our gratitude for your interest and commendable effort in comprehending the proposed theme and purpose. Your invaluable recommendations and suggestions have significantly enhanced our work and we sincerely appreciate your contribution. Thank you for taking the time to provide us with your insights.

GENERAL COMMENTS

Point 1.  Recommendation: In the abstract (lines 17-8) it is written: “from all 40 Picea abies (L.) Karst trees identified in the study area”. In Fig. 1b only trees with diameter higher than 8.5-10 cm are illustrated? Does this mean that you measured/ took into account only trees with D > 8.5 or you did not have younger trees in the selected study site? Please clarify.

Response 1. As stipulated in the abstract section, all Picea abies trees were identified and inventoried in the study area, a total of 40 trees. These trees were measured, resulting in diameters ranging from a minimum of 8.5 cm to a maximum of 92 cm. This information was included in the first paragraph of the Results section. Figure 1b. illustrates all the trees inventoried by GPS data collecting and mapping, and there are no other Picea abies trees younger or smaller than 8.5 cm in diameter.

Point 2. Recommendation: Part of the discussion section where the correlations with climate are presented should move to the Results Section.

Response 2. The findings of our dendrogemorphological analysis only pertain to the reconstruction of rockfall activity, as well as the results of correlations between them and the variations of meteorological parameters. The meteorological data was obtained from the ROCADA database and was only used in our analysis, and not represent the results of our measurements. We used this data to illustrate potential climatic influences on the dynamics of the analyzed geomorphological process, which validated our findings regarding the rockfall activity. Based on this validation process, we believe that the correlations between the rockfall activity and the meteorological parameters should be included in the Discussion section.

Point 3. Recommendation: In different parts of the manuscript you use the term “global climate change”. Global is not necessary. “Climate change” is enough.

Response 3. Accepted: The word “global” from “global climate change” was erased from the text.

Point 4. Recommendation: Sometimes you use rockfall(s) as one word and others as two words. Please homogenize.

Response 4. Accepted: The term “rockfall” is used as a single word.

Point 5. Recommendation: I think a better connection with the scope of the special issue is needed. You should try to emphasize the usefulness of your findings in terms of the landscape protection and/or the provided ecosystem services.

Response 5. Accepted. “The growth rings of trees can register environmental changes such as climate and geomorphological processes. These growth disturbances allow us to reconstruct rockfall activity over long periods. Our study highlights the potential of trees both for the reconstruction of rockfalls and the for evolution of tree characteristics such as number of trees, diameter, age, density, and spatial distribution.

In Figure 4, the spatial pattern of rockfalls is shown alongside the evolution of the tree features. The number of trees that have reached at least the sampling height which is 1.30 m, has increased unevenly from 4 trees in 1899 to 40 in 2021, with an average rate of 3 trees per 10 years. The maximum variation number was up to 25 trees in 1929, and the current number of trees (40) was reached in 1990. The structure of the forest stand in its current shape (number of trees, diameter, spatial distribution, including tree clusters) was completed in the 1960s. Subsequently, major changes only occurred in the diameters of the trees. Therefore, the same database obtained in the field can be used for both analyses on geomorphic processes and the evolution of forest landscapes.”

SPECIFIC COMMENTS

Point 1.  Recommendation: Line 17: identified or recorded?

Response 1.  Accepted: “identified”.

Point 2. Recommendation: Line 28, lines 33-34: “to human-made infrastructures and activities in mountainous regions”. There is some repetition here. Please revise. Also, why to mention only the human made infrastructure? It also affects the natural environment.

Response 2. Accepted: Line 28:Rockfall is a natural hazard that can affect human-made infrastructure and activities in mountainous regions”.

Line 33-34:” Due to the increasing frequency and intensity of natural disasters caused by climate change, rockfall processes pose a growing risk on slopes”.

Point 3. Recommendation: Line 46: “or incomplete”. Better to replace it with “or are incomplete”.

Response 3. Accepted: […] “or are incomplete”.

Point 4. Recommendation: Line 54: the parenthesis is not necessary.

Response 4. Accepted.

Point 5. Recommendation: Lines 57: I do not understand what do you mean with “two directions of approach”. By reading a) and b). I would write something like this: “rockfall activity was studied in two directions.”

Response 5. The perspectives of the main objectives of our work have two different directions, the spatial and temporal dimensions of rockfall activity patterns over the two last centuries. The manuscript is structured in the same way as dendrogeomorphological analysis methods. We declare that we would prefer to maintain the actual version of the study structure for a better and deeper understanding of the analyzed process and for a rigor scientific reasoning.

Point 6. Recommendation: Line 74: “by precipitations which turn into rain or snow.” Otherwise what are the precipitations?

Response 6. Accepted. “During these periods, the thermal weathering of the rocks is the most intense, and often is enhanced by precipitations.”

Point 7. Recommendation: Line 88: Give the full scientific name for Fagus sylavtica as you did for Picea.

Response 7. Accepted. “[…] Fagus sylvatica (L.).”

Point 8.  Recommendation: Line 94: delete double “do not”.

Response 8.  Accepted. 

Point 9.  Recommendation: Lines 108, 122 etc: no need to repeat the full name. Please replace just with Picea abies. It is enough to give the full scientific name once at the beginning.

Response 9.  Accepted.

Point 10.  Recommendation: Figure 2: Explain in the figure caption all the abbreviations appearing in the figure. Figures should be self-explicit.

Response 10.  Accepted. “Figure 2. Growing disturbances (GDs) generated by rockfall events identified through sampled Picea abies (L.) Karst growth rings: a) SCs (impact scars), GSs (sudden suppression of growth rings), and CW (compression wood), b) TRDs (traumatic resin ducts), c) SCs and TRDs.”

Point 11. Recommendation: Lines 188-189: “into 457 were GS…” English revision is needed. All the “were” are not necessary.

Response 11.  Accepted.

Point 12. Recommendation: Lines 189192: English revision is needed. The meaning is unclear.

Response 12. We noticed that some of the 1137 GDs recorded are simultaneously located in the same tree growth ring or were recorded during the first 10 years of the tree’s life. Due to GDs simultaneity could have the potential to induce errors in the reconstituted rockfall activity by overestimation and, therefore we removed them from the dendrogeomorphological analysis. After dendrochronological dating, from the GDs total, they left 945 growth disturbances, which correspond to as many rockfall events.

Point 13. Recommendation: Line 262: “within survey area”. Add “the” before survey.

Response 13.  Accepted. “[…] within the survey area […]”

Point 14. Recommendation: Lines 276-279: This is a common problem when dealing with the reconstruction of past events/disturbances, (e.g. fire history reconstruction etc). I would like to see a short paragraph about this in the Discussion section.

Response 14. The local historical archives reveal no records of major events (e.g. fire, epidemics, forest exploitation) that would have disturbed the analyzed forest stand. Thus, the idea from which our study started was that a geomorphic natural process shaped the forest landscape in the past two centuries, and this was mainly rockfall. However, we design in the near future an interdisciplinary study on the influence of other natural and anthropogenic factors on rockfall activity that could have hypothetically affected the forest stand in the past, except the rockfall process.

Point 15. Recommendation: Line 291: “for time interval”. Add “the”.

Response 15. Accepted. “[…] for the time interval […]”

Point 16. Recommendation: Line 293: “we taking in account”. English revision is needed.

Response 16. “we taking in account” is replaces with “consider”. “[…] if we consider the threshold of only 5% of CIP […]”.

Point 17. Recommendation: Line 368: are necessary.

Response 17. Accepted: “[…] the following steps are necessary:”

Comments on the Quality of English Language

Only minor English editing is required

Response 18. Accepted. The recommendation was followed and the minor English editing request was fulfilled.

Reviewer 2 Report

Comments and Suggestions for Authors

Dear Authors,

Thank you for your professional and insightful paper. It's complex, but interesting and valuable work, that need to be continued and further elaborated for more improved and representative results. I have some observations below that, in my view, could further improve the article:

Lines 80-81 – in one case the size of debris is in diameter (0,15m- 80th line), but in other case it is in m3 (0,3m3)- The consistency in presenting of units is important.

Line 91-92 – the judgment “…there may have been limited forestry or biological interventions in the past century, although there is no evidence to support this” should be supported by any reference that covers the study area (or that part of Southern Carpathian)

Line 94- “do not” is repeated.

-Lines between 335-349 should be moved to introduction as content vise, but also to cover the literature analyses on rockfall frequencies and meteorological data link.

Also, I think that the link between rockfalls with seismic activities, that are triggering rockfalls should be properly addressed. 

Hydrometeorological data are not presented and analyzed sufficiently in the sections “Materials and methods” and “Results”, instead presented in detail in “Discussions”. 

Author Response

We, the authors of this manuscript, express our gratitude for your interest and commendable effort in comprehending the proposed theme and purpose. Your invaluable recommendations and suggestions have significantly enhanced our work and we sincerely appreciate your contribution. Thank you for taking the time to provide us with your insights.

COMMENTS and SUGGESTIONS:

Point 1. Recommendation: Lines 80-81 – in one case the size of debris is in diameter (0,15m- 80th line), but in other case it is in m3 (0,3m3)- The consistency in presenting of units is important.

Response 1. Accepted: “[…] fragments smaller than 0.15 m in diameter, while the lower sector has fragments of approximately 0.7 m in diameter.”

Point 2. Recommendation: Line 91-92 – the judgment “…there may have been limited forestry or biological interventions in the past century, although there is no evidence to support this” should be supported by any reference that covers the study area (or that part of Southern Carpathian).

Response 2. Accepted.

Point 3. Recommendation: Line 94- “do not” is repeated.

Response 3. Accepted.

Point 4. Recommendation: Lines between 335-349 should be moved to introduction as content vise, but also to cover the literature analyses on rockfall frequencies and meteorological data link.  

and

Point 6. Recommendation: Hydrometeorological data are not presented and analyzed sufficiently in the sections “Materials and methods” and “Results”, instead presented in detail in “Discussions”. 

Response 4. and Response 6. The findings of our dendrogemorphological analysis only pertain to the reconstruction of rockfall activity, as well as the results of correlations between them and the variations of meteorological parameters. The meteorological data was obtained from the ROCADA database and was only used in our analysis, and not represent the results of our measurements. We used this data to illustrate potential climatic influences on the dynamics of the analyzed geomorphological process, which validated our findings regarding the rockfall activity. Based on this validation process, we believe that the correlations between the rockfall activity and the meteorological parameters should be included in the Discussion section.

Point 5. Recommendation: Also, I think that the link between rockfalls with seismic activities, that are triggering rockfalls should be properly addressed. 

Response 5. We design in the near future an interdisciplinary study on the influence of other natural and anthropogenic factors on rockfall activity, seismic hazard included, that could had hypothetically affected the forest stand in the past.

Reviewer 3 Report

Comments and Suggestions for Authors

This research focuses on the dendrogeomorphological reconstruction of rockfall activity in a forest stand. Rockfall activity, intensified by global climate change, poses a growing risk to human-made infrastructure in mountainous regions. Predicting these geomorphological processes is challenging due to spatiotemporal variations, including hazard zonation and associated risk assessment. Monitoring and understanding rockfall patterns are crucial for mitigating damages. This article explores methods to overcome data gaps for assessing rockfall activity magnitude over centuries and makes a significant contribution to understanding rockfalls through a dendrogeomorphological approach.

The approach is appropriate and constitutes a strong point. In this context, I am absolutely excited to learn from this article the main trajectory of falling rock fragments and temporal variations in rockfall activity magnitude. The results have important practical implications, but some clarifications and additional developments could enhance the study's robustness.

In this manuscript, the bibliographical references are in part well chosen, but the introduction deserves to be improved to better contextualize the theme. The section on climate change in the discussion should be presented first in the introduction

Some sections in the discussion deserve to be presented in terms of results: The correlation between the CIP-corrected rockfall reconstruction and meteorological parameters.  These results can then be discussed in the descussion section.

The discussion section should be reworked to improve the quality of the manuscript.

Author Response

We, the authors of this manuscript, express our gratitude for your interest and commendable effort in comprehending the proposed theme and purpose. Your invaluable recommendations and suggestions have significantly enhanced our work and we sincerely appreciate your contribution. Thank you for taking the time to provide us with your insights.

COMMENTS and SUGGESTIONS:

Point 1. The approach is appropriate and constitutes a strong point. In this context, I am absolutely excited to learn from this article the main trajectory of falling rock fragments and temporal variations in rockfall activity magnitude. The results have important practical implications, but some clarifications and additional developments could enhance the study's robustness.

Recommendation: In this manuscript, the bibliographical references are in part well chosen, but the introduction deserves to be improved to better contextualize the theme. The section on climate change in the discussion should be presented first in the introduction. Some sections in the discussion deserve to be presented in terms of results: The correlation between the CIP-corrected rockfall reconstruction and meteorological parameters.  These results can then be discussed in the descussion section.

Response 1. The findings of our dendrogemorphological analysis only pertain to the reconstruction of rockfall activity, as well as the results of correlations between them and the variations of meteorological parameters. The meteorological data was obtained from the ROCADA database and was only used in our analysis, and not represent the results of our measurements. We used this data to illustrate potential climatic influences on the dynamics of the analyzed geomorphological process, which validated our findings regarding the rockfall activity. Based on this validation process, we believe that the correlations between the rockfall activity and the meteorological parameters should be included in the Discussion section.

Point 2. Recommendation: The discussion section should be reworked to improve the quality of the manuscript.

Response 2. We believe that the Discussion section is improved through the next considerations regarding the forest landscape:

The growth rings of trees can register environmental changes such as climate and geomorphological processes. These growth disturbances allow us to reconstruct rockfall activity over long periods. Our study highlights the potential of trees both for the reconstruction of rockfalls and the for evolution of tree characteristics such as number of trees, diameter, age, density, and spatial distribution.

In Figure 4, the spatial pattern of rockfalls is shown alongside the evolution of the tree features. The number of trees that have reached at least the sampling height which is 1.30 m, has increased unevenly from 4 trees in 1899 to 40 in 2021, with an average rate of 3 trees per 10 years. The maximum variation number was up to 25 trees in 1929, and the current number of trees (40) was reached in 1990. The structure of the forest stand in its current shape (number of trees, diameter, spatial distribution, including tree clusters) was completed in the 1960s. Subsequently, major changes only occurred in the diameters of the trees. Therefore, the same database obtained in the field can be used for both analyses on geomorphic processes and on the evolution of forest landscapes.”